# OpenReview forum: "Understanding Curriculum Learning in Policy Optimization for Online Combinatorial Optimization"
_ICLR.cc/2023/Conference — Submitted to ICLR 2023_

### Official Review · Reviewer_GcdD · 2022-10-21

**Confidence:** 2
**Correctness:** 3
**Technical Novelty And Significance:** 3
**Empirical Novelty And Significance:** 3
**Recommendation:** 6

**Clarity, Quality, Novelty And Reproducibility:**

The clarity, quality, and reproducibility are good.

The novelty seems good, but the theoretical contributions were thoroughly examined. In addition, the paper could better explain the technical challenges in applying the existing frameworks.


**Strength And Weaknesses:**

Strength:

- S1: A combination of latent Markov Decision process, natural policy gradient technique, and curriculum learning can be an effective solution to online combinatorial optimization problems, and therefore, its theoretical foundations are important.
- S2: The presented analysis seems sound, but the proofs are not completely checked.
- S3: This paper is well-written, with formal settings being provided. Many details are carefully organized in the supplementary materials.


Weaknesses:

- W1: The so-called latent MDP appears to be a linear combination of classic MDPs. In such a sense, the existing analysis of MDP might be applicable to latent MDP without many technical challenges. The paper could better explain the technical challenge in obtaining the theoretical results.
- W2: Some minor comments:
     - While the main contribution of this paper is on the theoretical side, the experiments could be improved by having more baselines and problems. In particular, given the simple nature of the two considered problems, the paper may include simpler baselines based on, for example, multi-armed bandits. In addition to the interest of  Theorem 7, is there any other justification for the choice of the two selected problems?
  - Is the entropy regularization equivalent to enforcing a certain prior distribution?
  - The burden of notations might be reduced by avoiding introducing new notations, e.g., V^{\pi}=V^{\pi,0}, F(\theta)=\sum_{d^{\theta}}^{\theta}, and “here /club can be any symbol”.
  - What is the logic behind sampling from a fixed policy rather than the current policy?
  - The concept of environment E could be treated in a formal manner.


**Summary Of The Paper:**

This proposes to solve online combinatorial optimization problems using reinforcement learning composed of latent MDPs and curriculum learning. The paper presents formal formulations and theoretical results regarding the performance bound based on the relative condition number. In addition, the paper also presents theoretical results showing the power of curriculum learning in reducing the relative condition number of the selection (secretary) problem. Experimental studies are also provided to support the theoretical findings.


**Summary Of The Review:**

This paper proposes to solve online combinatorial optimization problems using reinforcement learning composed of latent MDPs and curriculum learning, and presents theoretical studies as well as experimental results. This paper is overall good and sound, but the significance of the theoretical contribution needs better justifications.

---

> ### Author Response · Authors · 2022-11-05
> **Response to Reviewer GcdD**
>
> Thank you for your review!
>
> **W1:** Indeed, in the policy optimization setting, the algorithm of classic MDPs can be nearly directly applied to LMDPs, because no context (which MDP in the LMDP is the agent interacting with) is explicitly needed. The technical challenge is present in the online learning (e.g., tabular RL) setting, where the context is important and LMDPs do not have the basic Bellman optimality as MDPs do. But this is irrelevant to this paper.
>
> **W2**:
> - The purpose of this paper is to show that **curriculum learning speeds up training because it finds a better sampling policy**. By adding more baselines, we would need to compare different training techniques as well as different curriculum designs, which we believe does not help much to the justification of our simple purpose. We agree that the experiments can be further completed if more problems are added. **We don't think multi-armed bandits, if not MDPs, can model online CO problems adequately, and this is the reason we choose LMDPs.** The reason we choose these two problems is stated at the beginning of Section 3.
>
> - In my opinion, entropy regularization simply avoids a deterministic policy which could be local-optimum.
>
> - Thank you for your advice. We will improve the notations in the revision.
>
> - This is for the convenience of theoretical analyses. We cannot calculate $\kappa$ when the sampling policy is changing. In experiments, we draw the curve of the current policy.
>
> - Yes we agree with this point. We will improve in the revision.

---

### Official Review · Reviewer_qByH · 2022-10-24

**Confidence:** 1
**Correctness:** 3
**Technical Novelty And Significance:** 3
**Empirical Novelty And Significance:** 3
**Recommendation:** 5

**Clarity, Quality, Novelty And Reproducibility:**


<Methodology>

1. Does the proposed methodology suggest a particular representation or learning method for dealing with combinatorial structures?

2. What is the reason for using a natural gradient instead of a typical gradient-based optimization method? Can you describe the advantages of using a natural gradient in terms of computational complexity and convergence speed?

3. In curriculum learning, how to quantify the difficulty of the task?

4. How does the convergence rate change when the distribution for multi-task changes?



<Experiments>

The proposed algorithm does not compare its performance with other baseline algorithms. The provided experiment results are nothing more than an ablation study that proves the effectiveness of curriculum learning.


**Strength And Weaknesses:**


This study formulates online combinatorial optimization and proposes a way to derive optimum sequential decision-making policy. In addition, the current study provides a rigorous analysis of the performance of the policy.

I think the level of analysis and mathematical rigor is very high. Since this field is not my area of expertise, I could not accurately evaluate this process.

I think the current study fits better with an optimization or statistical analysis paper rather than a machine learning conference. I believe the current study should at least provide intuition or inspiration on how to effectively employ a learning-based solver for such a problem based on the discussed mathematical properties and structures.


**Summary Of The Paper:**

The current study formulates online CO problems as latent Markov Decision Processes and propose a policy optimization method using natural policy gradient. In addition, the current study characterizes performance and investigates the effect of curriculum learning in deriving optimum policy.

**Summary Of The Review:**

I think the level of analysis and the mathematical rigor are very high. Since this is not my area of ​​expertise, I could not accurately evaluate this course. However, I think this paper is closer to optimization or optimal control paper than a machine learning paper.

---

> ### Author Response · Authors · 2022-11-05
> **Response to Reviewer qByH**
>
> Thank you for your review!
>
> **Methodology**
> 1. Our methodology, Natural Policy Gradient combined with curriculum learning, does not impose an explicit demand on the representation of states (i.e., the feature representation $\phi (s, a)$ implicitly affects the value $\epsilon_{\text{bias}}$), but as we mentioned in Appendix C.1, a "scale-invariant" representation is helpful for generalization into larger problems. Our methodology underscores curriculum learning as a simple yet efficient method for dealing with combinatorial structures.
>
> 2. NPG is faster than Policy Gradient methods (https://arxiv.org/pdf/2102.11270.pdf). For our specific choice of log-linear policy, computing natural gradient takes $O (d^2)$ time (because need to compute matrix inverse), since $d = 20$ is not large, this is not much than the $O(d)$ complexity of gradient. But as our theory suggests, NPG has linear convergence, which is much faster than the $O(1 / \sqrt{T})$ convergence of PG with log-barrier regularization [Agarwal et al., 2021].
>
> 3. We focus on online Combinatorial Optimization problems, so the difficulty is quantified by the scale of the problem. We think a Secretary Problem with $100$ candidates is harder than that with $10$ candidates.
>
> 4. This is hard to say. Our result is an upper bound, which means whatever the distribution for multi-task changes to, the convergence is always dominated by an exponentially shrinking leading term.
>
> **Experiments:** The purpose of this paper is to show that **curriculum learning speeds up training because it finds a better sampling policy**. Results in Section 6.2 indeed do this because they show that **a very simple curriculum (a smaller-scaled problem, even without carefully designed instance distribution) is sufficient in many cases**. If to go beyond the naive random sampling policy, we would need to compare different training techniques as well as different curriculum designs, which we believe does not help much to the justification of our simple purpose.

---

### Official Review · Reviewer_eGuH · 2022-10-24

**Confidence:** 3
**Correctness:** 4
**Technical Novelty And Significance:** 2
**Empirical Novelty And Significance:** Not applicable
**Recommendation:** 5

**Clarity, Quality, Novelty And Reproducibility:**

The paper has some misleading claim in the abstract about reducing the relative condition number exponentially by applying curriculum learning. It's only for the SP problem and w.r.t. the naive random sampling policy.
I also suggest the author/s considering moving the related work section to the end, because it has some notations not defined before.
For the novelty part, please check my comments above.

**Strength And Weaknesses:**

Strength:
  1. The paper reformulates the combinatorial optimization as a latent markov decision process (LMDP), which is interesting and useful in applying the technics of the RL in LMDP case.
  2. The paper's motivation to find when and why the curriculum learning is helpful in applying RL in LMDP is meaningful. It also shows that the curriculum learning is helpful in finding a strong sampling policy to reduce the distribution shift by either formally proving or empirically showing that the relative condition number is reduced.

Weakness:
  1. The technics behind the theoretical results on the convergence rate of the proposed algorithm are well-known, which makes the theoretical result like Theorem 6 very incremental
  2. The theoretical result to show that the curriculum learning could find a good sampling policy and reduce the relative condition number exponentially is only w.r.t. the naive random sampling policy, which is not usually used in practice.

**Summary Of The Paper:**

The paper investigates the impact of the curriculum learning used in applying reinforcement learning to solve the combinatorial optimization problems. To demonstrate the impact of the curriculum learning, the paper first formulates the combinatorial optimization problem as latent markov decision process (LMDP) and applying the technics developed in LMDP to show the convergence rate. Besides that, the paper shows that the curriculum learning is helpful in finding a strong sampling policy to reduce the distribution shift. It also formally proves that the relative condition number is reduced exponentially with the curriculum learning comparing to the naive random sampling policy.

**Summary Of The Review:**

The paper focuses on finding the role of curriculum learning in helping the RL algorithms to solve the CO problems, which is well motivated and interesting. The paper has a few misleading notations but overall the paper is okay to read and understand. The insights of treating CO problems as LMDP is interesting but somehow straightforward. The convergence rate theoretical result is very incremental comparing with previous RL works in LMDP domain. For the curriculum learning role in the RL algorithms for solving the CO, it's interesting to see that it help reduce the relative condition number when using it as sampling policy, but the baseline is only the naive sampler which is not ideal when considering the practical usage.

---

> ### Author Response · Authors · 2022-11-04
> **Response to Reviewer eGuH**
>
> Thank you for your review!
>
> **Weakness 1:**
> Our result of "NPG for LMDP" (Algorithm 3, Theorem 6, and Lemma 19) **covers most practical situations: sample-based, with entropy regularization and batched updates of weight (line 14 of Algorithm 3)**. By the time of finish, no previous work provided a unified analysis for these situations to the best of our knowledge. Also, the use of batched updates instead of successive projected gradient ascent (as [Agarwal et al., 2021]) is often used in practical training. So we believe our convergence result is not very limited.
>
> **Weakness 2:**
> The purpose of this paper is to show that **curriculum learning speeds up training because it finds a better sampling policy**. Results in Section 6.2 indeed do this because they show that **a very simple curriculum (a smaller-scaled problem, even without carefully designed instance distribution) is sufficient in many cases**. If to go beyond the naive random sampling policy, we would need to compare different training techniques as well as different curriculum designs, which we believe does not help much to the justification of our simple purpose.
>
> **Clarity:**
> The drastic decrease in $\kappa$ is also corroborated by experiments in Online Knapsack problems, please refer to Figure 2, 7, and 8. In sub-figure (b), the $y$-axis is $\log (\kappa)$, so a constant gap in $y$ coordinate means an exponential gap in $\kappa$.

---

### Official Review · Reviewer_1B7j · 2022-10-30

**Confidence:** 3
**Correctness:** 3
**Technical Novelty And Significance:** 2
**Empirical Novelty And Significance:** 1
**Recommendation:** 3

**Clarity, Quality, Novelty And Reproducibility:**

The paper meanders through algorithmic approaches with little insight into the performance they offer in the results.  To be true, I cannot follow how the algorithm described in Section 4 fits into a conventional RL approach.

Aside from the question of how curriculums are defined, What exactly does "history-independent" mean? Is this about knowledge of the state of previous stages?  In MDP's an optimal policy only need depend on the current state, unlike when the state is unobservable.  What then is the condition for LMDPs?

A few words that describe the relevance of using Natural Policy Gradients would also help clarify the  paper's approach.

**Details Of Ethics Concerns:**

Just a small point -- common usage is and has been to refer to the "secretary problem" by other names that use less culturally loaded terms.

**Strength And Weaknesses:**

The memorable take-away from the paper is the use of "Latent Markov Decision Process (LMDP)" to study sequential decision problems where the state cannot be fully observed, as opposed to trying to express these as partially  observable MDPS (POMDPs.) Solutions to  no-recall GAP problems is a vital area and deserves study.

Curriculum implies exploiting a heuristic about problem complexity - to relate an evolution of simplicity versus complexity during training.  The only description of the curriculum I find is in the description of experimental results, that "There is no explicit relation between the curriculum and the target environment, so the curriculum can be viewed as random and independent. "  So as stated, "even if the curriculum is randomly generated" shouldn't this be viewed as a kind of random-restarts approach?

There is also the question of reframing GAP problems as RL-style problems that either implicitly or explicitly consider the probability distribution of samples. In related work "Currently, Online Matching and Online Knapsack have only approximation algorithms (Huang et al., 2019; Albers et al., 2021)."    This is true, but "online matching -- eg. the SP -- is underspecified as a probability model. These "generalized assignment problems" (Albers 2021) do not consider the probability distribution of samples.  In short, one needs to make an assumption about the sample, as the authors do,  that the sample distributions are relevant; value and size distributions were assumed uniform (Section 3.2).   As Granger ("Optimal Statistical Decisions, 1970, p. 331) states " In the context of this fantasy (that only information received is the rank of the sample) it is assumed at the beginning of the process (one) knows nothing about the quality" (e.g the distribution they are drawn from) of the n items one will see.   One may argue this is unrealistic, more importantly it obscures much of what's going on in a sequential process.

As a minor point, for the statement  "As surveyed in Bengio et al. (2009), Curriculum Learning has been applied to training deep neural networks and non-convex optimizations and improves the convergence in several cases." I think Bengio (2020) is the citation of the survey you mean. DNNs were not prevalent in 2009, and the 2009 paper works merely with a 3-level NN.


**Summary Of The Paper:**

This paper presents an exploration of the combination of curriculum learning for on-line reinforcement learning problems by application to two related combinatoric optimization problems.  Central to the approach is to solve for a mixture $w_m$ of MDP models called a "Latent Markov Decision Process (LMDP) (Kwon et al., 2021a)" by the relative condition number κ is reduced.

Curriculum learning, as popularized in Bengio (2009), presents an approach to non-convex optimization in learning problems, that takes advantage of training on progressively more challenging subsets of cases. Insights from the domain inform what makes cases  more challenging. This has a regularization effect, in addition to guiding - or "shaping" the optimization toward desired solutions.

Online combinatoric problems, that have been termed "generalized assignment problems" (GAP) (Albers 2021), attempt to rank items where the policy is restricted to making a selection "without recall" -- once an item is passed over, it may not be chosen.  These belong to a class of optimal stopping sequential decision problems.

**Summary Of The Review:**

It's a credit to the authors to have made realistic claims, and to have provided valid evidence to support them.  Granted that the work is correct, the results are not substantial.  Again one recognises as stated that this is exploratory work on a novel formulation.  I would question how the problem is approached, and wonder if as stated it does not admit of a clean solution.

---

> ### Author Response · Authors · 2022-11-06
> **Response to Reviewer 1B7j**
>
> Thank you for your review!
>
> ### Description of the curriculum:
> Kong et al. [2019] deployed a sequence of gradually harder problems as a curriculum. In fact, they studied the classical Secretary Problem (or the best choice problem if the name is considered discriminative). As a result **their curriculum consisted only of the classical ones** (but with smaller problem scales).
>
> However, we consider the cases where **the curriculum and the target problem have different instance distributions**, which is a generalization of and a complement to their approach. We say "There is no explicit relationship between the curriculum and the target environment, so the curriculum can be viewed as random and independent.", because **we did not deliberately design the curriculum**.
>
> Take the SP as an example, we set up a procedure to parameterize the generation of the instance distribution, given a random seed. All the experiments (other than the classical one) were run with different random seeds for the one-step curriculum and the target problem. Since the random seeds are different, the random processes for the two phases are independent. We **cannot draw any explicit relation between the instance distributions in the curriculum and the target environment**. We'll make this point clearer in the revision.
>
> Section 6.2, Section 7, and Appendix C show that a one-step, random curriculum is sufficient to boost the convergence by reducing $\kappa$ exponentially. But **a random curriculum cannot be simply said to be a kind of random-restarts approach**. This is because the **core idea in curriculum learning for online CO problems is to use the same type of problem in the curriculum**. Not using the same distribution as the target environment (which is also not possible to be known) is OK. This is a highly structured initialization. We mention the random generation to emphasize the simplicity and efficacy of our curriculum learning approach.
>
> ### Probability distribution:
> I think this is a difference of perspectives we have towards solving these problems. Our choice of solving these problems under a distribution over instances is a Machine Learning style instead of a Theoretical Computer Science style. The former usually optimizes over a data set that inherently **requires average performance**, while the latter usually studies the worst case guarantee which is **minimax performance**. Our ML-type idea is also built upon the nature of the Secretary Problem, because the ordering of candidates is not known, and **there is no algorithm that guarantees a non-zero worst-case performance on each ordering**. Again, many of the ML/RL for CO works mentioned in Sections 1 and 2 used this **average performance** setting, whose distributions were parameterized by some algorithm. So we believe in the field of ML, this is a natural choice to tackle GAP problems and any other CO problems.
>
> ### DNN:
> Thanks for pointing it out. The 2009 paper did experiments on 3-layer NNs and claimed the efficacy of curriculum learning applied to DNNs. The 2020 paper, however, did not focus on curriculum learning, so we think it is irrelevant to this part of related works.
>
> If there are questions not (properly) answered, please feel free to reply!

---

### Decision · Program_Chairs · 2023-01-20

**Decision:**

Reject

**Justification For Why Not Higher Score:**

Thanks to the author responses, some issues raided in the initial reviews were clarified. However, there are still issues that need to be further investigated, such as the insight into the performance of the proposed approach and quantifying the difficulty of the task in curriculum learning. Overall, the contributions brought by this paper are not substantial and the novelty of theoretical analysis is limited. Therefore, this paper cannot be recommended for acceptance.


**Justification For Why Not Lower Score:**

N/A

**Metareview: Summary, Strengths And Weaknesses:**

Summary:
This paper studies policy optimization for online combinatorial optimization. The authors formulated online combinatorial optimization problems as latent Markov Decision Processes and proved convergence bounds for natural policy gradient methods. Furthermore, it is shown that curriculum learning helps find a strong sampling policy and reduce the distribution shift. Experiments show the validity of the proposed method.

Strengths:
Combining latent Markov Decision processes, natural policy gradient techniques, and curriculum learning is interesting and shown to be effective effective. The paper is overall well-written and sound.

Weaknesses:
Thanks to the author responses, some issues raided in the initial reviews were clarified. However, there are still  issues that need to be further investigated, such as the insight into the performance of the proposed approach and quantifying the difficulty of the task in curriculum learning. Also, the theoretical analysis has limited novelty.